# Thermal Stability, Hardness, and Corrosion Behavior of the Nickel–Ruthenium–Phosphorus Sputtering Coatings

**Yu-Cheng Hsiao and Fan-Bean Wu \***

Department of Materials Science and Engineering, National United University, Miaoli 36003, Taiwan; fanbeanwu@gmail.com
\* Correspondence: fbwu@nuu.edu.tw; Tel.: +886-37-382232

**Abstract:** Nickel–ruthenium–phosphorus, Ni–Ru–P, alloy coatings were fabricated by magnetron dual-gun co-sputtering from Ni–P alloy and Ru source targets. The composition variation and related microstructure evolution of the coatings were manipulated by the input power modulation. The as-prepared Ni–Ru–P alloy coatings with a Ru content less than 12.2 at.% are amorphous/nanocrystalline, while that with a high Ru content of 52.7 at.% shows a feature of crystallized Ni, Ru, and $Ru_2P$ mixed phases in the as-deposited state. The crystallized phases for high Ru content Ni–Ru–P coatings are stable against annealing process up to 600 °C. By contrast, the amorphous/nanocrystalline Ni–Ru–P thin films withstand a heat-treated temperature up to 475 °C and then transform into Ni(Ru) and $Ni_xP_y$ crystallized phases at an annealing temperature over 500 °C. The surface hardness of the Ni–Ru–P films ranges from 7.2 to 12.1 GPa and increases with the Ru content and the annealing temperatures. A highest surface hardness is found for the 550 °C annealed Ni–Ru–P with a high Ru content of 52.7 at.%. The $E_{corr}$ values of the heat-treated amorphous/nanocrystalline Ni–Ru–P coatings become more negative, while with a high Ru content over 27.3 at.% the Ni–Ru–P films show more negative $E_{corr}$ values after annealing process. The pitting corrosion feature is observed for the amorphous/nanocrystalline Ni–Ru–P coatings when tested in a 3.5M NaCl solution. Severer pitting corrosion is found for the 550 °C annealed Ni–Ru–P coatings. The development of Ni(Ru) and $Ni_xP_y$ crystallized phases during annealing is responsible for the degeneration of corrosion resistance.

**Keywords:** Ni–Ru–P; sputtering; annealing; corrosion

## 1. Introduction

Nickel–phosphorus (Ni–P) coating has been frequently adopted as a hard alloy coating due to corrosion resistance, high hardness and toughness, wear resistance, etc. [1–6]. The Ni–P alloy coatings are frequently deposited through wet chemical methods, including electroplating or electroless plating. Generally speaking, the electroplated or electroless Ni–P coatings possess a supersaturated microstructure feature in as-deposited and can be strengthened by the crystallization of Ni and the precipitation of Ni–P crystallites with appropriate heat treatment. However, the low crystallization temperature of approximately 350 °C of the Ni–P coatings is critical for high-temperature applications and further fields of use [7]. The introduction of the third elements and compounds, including Al [8], Cr [9], Mo [10], Cu [11–13], $Al_2O_3$ [14,15], CNTs [16,17], etc., are frequently proposed to further enhance its mechanical, thermal, and chemical properties. For instance, the thermal stability of the Ni–P layer could be improved by the introduction of a third element with high melting point like W [18,19]. The addition of W atoms, even at a low level of 3 at.%, has a significant effect on promoting the

thermal stability of the ternary Ni–W–P films [19]. The precipitation of $Ni_3P$ phase, which takes place at around 350 °C in a binary Ni–P alloy films, is postponed up to 450 °C due to W incorporation. The codeposition of W atoms also induce the W solid-solutioning in Ni crystalline phase, i.e., Ni(W) matrix, which further boosts the surface hardness over 10 GPa. The increase in thermal stability and mechanical properties of the binary Ni–P layer can be achieved by the addition of Al element to form a ternary Ni–Al–P film [8]. A two-stage strengthening by $Ni_3P$ and $Ni_xAl_y$ precipitations at 400 and 500 °C, respectively, is manifested. The hardness of the Ni–Al–P films is pushed up from 7.0, 9.5, to 11.0 GPa as annealed at 400, 500, to 550 °C. Marzo and coworkers [20] produced Cr-coplated Ni–P film and found an enlarged polarization resistance for the Ni–P–Cr film. However, the coplating of ternary Ni–P-based layers needs complex solution design. Thus, an alternative method using physical vacuum deposition is proposed [8,12,19,21,22]. The co-deposition of a third element into Ni–P alloy through the sputtering technique has been successfully adopted in Ni–P-based alloy coatings with good composition control. Since the phase evolution of the Ni–P-based thin film is sensitive to addition element content and thermal history, the control on third element concentration and annealing process are thus critical to the related properties. To enhance the thermal stability and corrosion resistance, various kind of material systems have been applied into Ni–P-based coatings as described above. The Ru element has been studied as a promising catalyst for hydrogen evolution reaction due to its small overpotential as compared to commercial Pt material [23]. The RuCo dispersion on Ru film has been proved as an effective catalyst for the hydrogenation of $CO_2$ [24], meaning that Ru can be produced in a fashion of alloy film to provide specific functions. However, reports on the feasibility of introduction of noble metals into the Ni–P deposits, which are characterized by their high thermal and chemical stability, are limited. In present case, the ruthenium is introduced as the additive element to Ni–Ru–P alloy thin films based on its high melting point, excellent chemical inertness and thermal stability. The ternary Ni–Ru–P coating is fabricated by the co-sputtering technique with input power control on metallic targets as sputtering sources. The microstructure feature, thermal stability, hardness and elastic modulus, and corrosion resistance of the Ni–Ru–P coatings with various Ru contents are analyzed. The effect of phase transformation due to annealing on thermal stability, surface hardness, and corrosion characteristics are discussed.

## 2. Materials and Methods

The Ni–Ru–P coatings were deposited through magnetron dual-gun co-sputtering technique on AISI 420 stainless steel substrates. The $Ni_{75}P_{25}$ compound and pure Ru targets both of 50.8 and 6.25 mm in diameter and thickness, respectively, were employed as sputtering sources. Two guns with a radio frequency magnetron sputtering, RFMS, input were adopted. Prior to deposition, the vacuum chamber was evacuated down to $4.0 \times 10^{-4}$ Pa followed by the inlet of high purity argon gas to a working pressure of $3.7 \times 10^{-1}$ Pa. The targets were pre-sputtered for 10 min in order to clean the target surfaces followed by the deposition process. During sputtering, the input power for Ru was tuned from 15 to 100 W with a fixed power at 100 W on the Ni–P compound target. The deposition time and substrate temperature were set as 2 h and 200 °C, respectively. The thickness for all coatings was controlled around 1 μm for reliable characterizations. To evaluate the coating structural stability under thermal process, the coated samples were annealed from 350 to 600 °C in a vacuum environment at $4.0 \times 10^{-4}$ Pa level for 3 h and then were furnace cooled. The chemical composition of the Ni–Ru–P coatings was evaluated with a field-emission electron probe microanalyzer (FE-EPMA, JXA-iHP200F, JEOL, Tokyo, Japan). The phases of sputtered coatings in as-deposited and annealed states were analyzed by a conventional X-ray diffractometer (XRD, TTRAX III, Rigaku, Tokyo, Japan). The surface images of the coatings were observed with a field-emission scanning electron microscope (FE-SEM, JEOL, JSM-6700, Tokyo, Japan). The coating thickness was also measured from the cross-sectional SEM image. The surface hardness and elastic modulus of various Ni–Ru–P coatings were examined using a nano-indentation tester (Nano Hardness Tester, CSM Instrument, Needham Heights, MA, USA) equipped with a Berkovich three-faced pyramidal diamond indenter under a test load of 3 mN.

At least five indents were made, measured and averaged on a statistical basis. The electrochemical polarization curve was generated by an electrochemical workstation (Jiehan-5000, Jiehan, Taiwan) with a build-in software for corrosion potential and current density calculation. An electrochemical cell with the 3.5 M NaCl solution maintained at 25 °C was used. A Pt counter electrode and a saturated calomel electrode (SCE) reference electrode were employed. The samples were immersed in the testing solution for 30 min to stabilize the open circuit potential (OCP). The scanning range was set form −0.25 to +0.25 V according to the OCP. Coating morphologies after chemical attack were observed via FE-SEM, as mentioned above.

## 3. Results and Discussion

### 3.1. Microstructure and Thermal Stability

The sputtered ternary Ni–Ru–P coatings are fabricated by the dual-gun sputtering system. At a fixed power of 100 W for the $Ni_{75}P_{25}$ target, the Ru contents of the Ni–Ru–P coatings are from 3.3 to 52.7 at.% under an input power range from 15 to 100 W, as listed in Table 1. The Ni and P concentration decrease systematically with the contents of Ru element in the alloy coatings. The Ni to P ratio in each Ni–Ru–P coating is also calculated for reference. The Ni/P ratios are around 4.1–4.6, regardless of the variation in Ru content in the Ni–Ru–P films. This indicates the supplies of the Ni and P from the $Ni_{75}P_{25}$ target are stable during co-sputtering process.

**Table 1.** The deposition conditions and composition of the sputtered Ni–Ru–P coatings.

| Coating Designation | Input Power (W) | | Thickness (μm) | Composition (at.%) | | | Ni/P |
|---|---|---|---|---|---|---|---|
| | $Ni_{75}P_{25}$ | Ru | | Ni | P | Ru | |
| A, $Ni_{78.9}Ru_{3.3}P_{17.8}$ | 100 | 15 | 0.94 ± 0.08 | 78.9 ± 0.3 | 17.8 ± 0.3 | 3.3 ± 0.1 | 4.4 |
| B, $Ni_{72.0}Ru_{12.2}P_{15.8}$ | 100 | 25 | 0.94 ± 0.10 | 72.0 ± 0.2 | 15.8 ± 0.2 | 12.2 ± 0.2 | 4.6 |
| C, $Ni_{58.5}Ru_{27.3}P_{14.2}$ | 100 | 50 | 0.85 ± 0.11 | 58.5 ± 0.2 | 14.2 ± 0.1 | 27.3 ± 0.3 | 4.1 |
| D, $Ni_{38.6}Ru_{52.7}P_{8.7}$ | 100 | 100 | 1.30 ± 0.15 | 38.6 ± 0.9 | 8.7 ± 0.2 | 52.7 ± 0.8 | 4.4 |

The phase evolution of the Ni–Ru–P coatings under various annealing temperatures is investigated through X-ray diffraction technique to evaluate the thermal stability of the alloy thin films. As plotted in Figure 1, for coatings with Ru contents less than 27.3 at.%, i.e., A, B, and C, the 200 °C as-deposited films show an amorphous/nanocrystalline feature, which is characterized by a representative broadened peak. The 2θ peak position is around 44.6°, where a Ni (111) orientation is suggested. Since Ni, P and Ru are co-sputtered, an amorphous/nanocrystaline Ni matrix with Ru and P incorporated microstructure is expected. With the increase of annealing temperature, the amorphous/nanocrystaline microstructure develops into the Ni and $Ni_xP_y$ phases, including $Ni_3P$, $Ni_5P_2$, and $Ni_{12}P_5$, as shown in Figure 1. It is noted that there is no Ru related diffraction peaks found in the patterns. In recent research of Shin [25], Ru–P alloy films remained an amorphous feature under vacuum annealing up to 500 °C. As a consequence, the Ni(Ru) with $Ni_xP_y$ intermetallic phases are confirmed for the $Ni_{72.0}Ru_{12.2}P_{15.8}$ coatings.

Discrepancy in phase transformation phenomenon is found for the Ni–Ru–P films with high Ru content of 52.7 at.%, as depicted in Figure 2. According to the phase diagrams [26], the Ni, Ru, and $Ru_2P$ are identified for the $Ni_{38.6}Ru_{52.7}P_{8.7}$ coatings under various annealing temperatures up to 600 °C. This implies the Ni, Ru, and $Ru_2P$ phases are thermally stable against heat treatments. It is argued that the multiphase feature which forms even under 200 °C as-deposited condition is attributed to the relative high power co-sputtering, i.e., 100 W for both targets. On the other hand, the P element depletes through the formation of $Ru_2P$ with Ru during film deposition. No further P can be used to form $Ni_xP_y$ compounds with Ni at elevated temperature during annealing. Stable and mixed Ni, Ru, and $Ru_2P$ phases are consequently observed for the $Ni_{38.6}Ru_{52.7}P_{8.7}$ coatings. Figure 3 shows the plane view FE-SEM images of the as-deposited B, $Ni_{72.0}Ru_{12.2}P_{15.8}$ and D, $Ni_{38.6}Ru_{52.7}P_{8.7}$ coatings. Figure 3b exhibits a smooth surface condition while significant crystallized morphology with many

tiny mounds is observed for the $Ni_{38.6}Ru_{52.7}P_{8.7}$ film. Such a difference is again attributed to the amorphous/nanocrystalline and mixed phase microstructures for low and high Ru content Ni–Ru–P coatings, respectively.

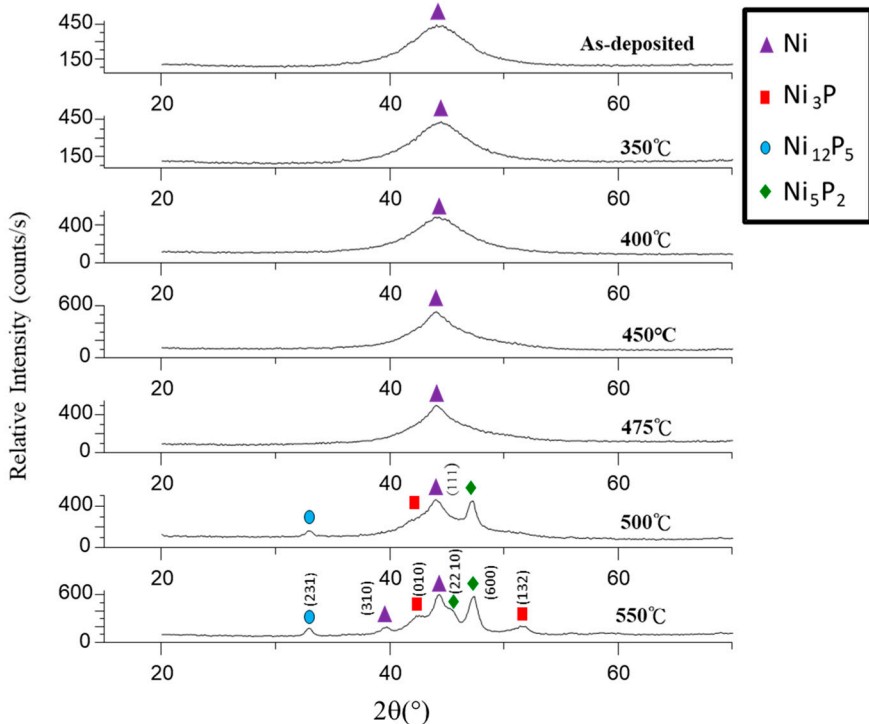

**Figure 1.** X-ray diffraction patterns of B, $Ni_{72.0}Ru_{12.2}P_{15.8}$ coatings annealed at various temperatures for 3 h.

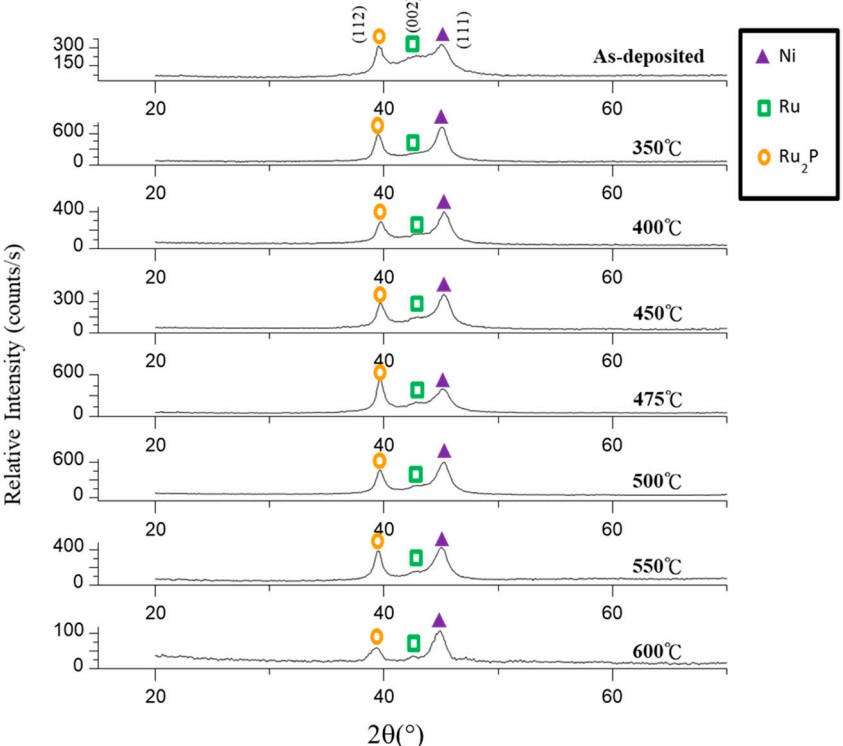

**Figure 2.** X-ray diffraction patterns of D, $Ni_{38.6}Ru_{52.7}P_{8.7}$ coatings annealed at various temperatures for 3 h.

## 3.2. Surface Hardness and Elastic Modulus

Table 2 depicts the surface hardness of various Ni–Ru–P coatings evaluated through nanoindentation technique. The hardness and elastic modulus for the as-fabricated and annealed samples are listed for comparison. For the as-deposited films, the surface hardness increases simply from 7.2 to 10.4 GPa with the addition of Ru from 3.3 to 52.7 at.%. Obviously the promotion in hardness is attributed by the Ru addition amounts. After heat treatment, the hardness of each film increases as compared to those in the as-deposited state. The hardness increases with annealing temperatures. For instance, the hardness of sample B, $Ni_{72.0}Ru_{12.2}P_{15.8}$, is enhanced from 7.2 to 8.9 and 9.5 GPa as annealing temperatures of 475 and 550 °C are applied, respectively. Sample C also shows a hardness increase with respect to temperature in the similar manner. The phase transformation from amorphous to nanocrystalline Ni matrix with $Ni_xP_y$ precipitation, as indicated in Figure 2, is the key to such strengthening. For the case of sample D, $Ni_{38.6}Ru_{52.7}P_{8.7}$, a mixed phase feature is generated in the as-deposited state, leading to a higher hardness value, as compared to those of samples A, B, and C. After annealing at 475 and 550 °C, the slight development of the mixed crystalline phases causes the increase from 10.4, to 11.1 and 12.1 GPa, respectively. It should be noted that in our previous research works on Ni–Al–P [8] and Ni–W–P coatings [7,19,21], the addition of the third elements, Al or W, is beneficial in promoting the thermal stability and hardness of the Ni–P-based ternary alloy coatings. The promotion is mainly due to the delay of Ni recrystallization and $Ni_3P$ precipitation toward higher temperatures. Moreover, the third elements, Al and W, dissolve into Ni crystalline phases and strengthen the Ni–P-based alloy coatings. In present case, it is argued that the addition of Ru below 27.3 at.% shows an identical strengthening effect on thermal stability and hardness of the Ni–Ru–P alloys. In addition, the elastic modulus shows the similar trend with composition and heat treatment process. For example, the as-deposited Ni–Ru–P films possess an elastic modulus lower than 190.9 GPa as Ru content is less than 27.2 at.%, while a highest elastic modulus of 210.3 GPa is measured for the $Ni_{38.6}Ru_{52.7}P_{8.7}$ film. With the heat treatments under 475 and 550 °C, the phase evolution of the Ni–Ru–P coatings is responsible for the gradual increase in elastic modulus for each Ni–Ru–P deposit.

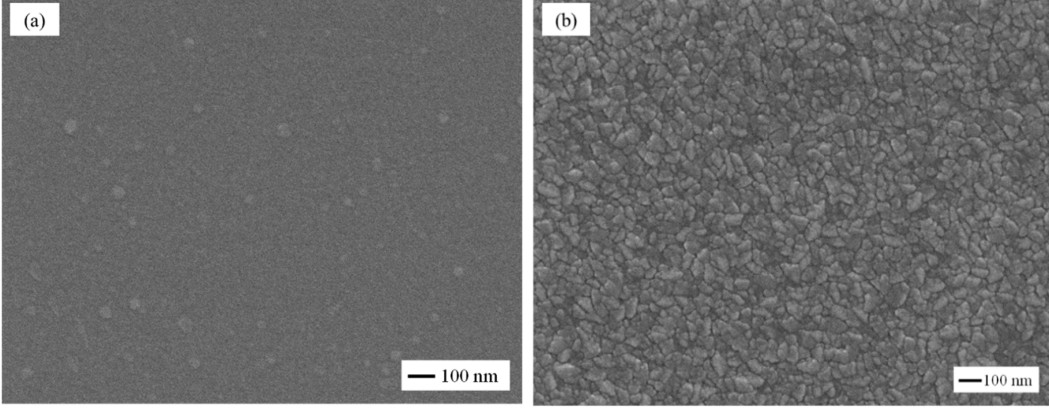

**Figure 3.** The plane view field-emission scanning electron microscope (FE-SEM) images of the as-deposited (**a**) B, $Ni_{72.0}Ru_{12.2}P_{15.8}$ and (**b**) D, $Ni_{38.6}Ru_{52.7}P_{8.7}$ coatings.

**Table 2.** The surface hardness and elastic modulus of the Ni–Ru–P coatings in as-deposited and annealed states.

| Coating Systems | Process Condition | Hardness (GPa) | Elastic Modulus (GPa) |
|---|---|---|---|
| A, $Ni_{78.9}Ru_{3.3}P_{17.8}$ | 200 °C as-deposited | 7.2 ± 0.1 | 161.8 ± 9.5 |
| B, $Ni_{72.0}Ru_{12.2}P_{15.8}$ | 200 °C as-deposited | 7.2 ± 0.4 | 138.3 ± 6.6 |
| | 475 °C annealed | 8.9 ± 0.5 | 164.9 ± 5.8 |
| | 550 °C annealed | 9.5 ± 0.2 | 201.1 ± 11.1 |

**Table 2.** *Cont.*

| Coating Systems | Process Condition | Hardness (GPa) | Elastic Modulus (GPa) |
|---|---|---|---|
| C, Ni$_{58.5}$Ru$_{27.3}$P$_{14.2}$ | 200 °C as-deposited | 8.1 ± 0.4 | 190.9 ± 0.4 |
| | 475 °C annealed | 9.1 ± 0.4 | 189.4 ± 11.3 |
| | 550 °C annealed | 9.5 ± 0.3 | 211.4 ± 8.4 |
| D, Ni$_{38.6}$Ru$_{52.7}$P$_{8.7}$ | 200 °C as-deposited | 10.4 ± 0.2 | 210.3 ± 1.6 |
| | 475 °C annealed | 11.1 ± 0.7 | 221.9 ± 14.7 |
| | 550 °C annealed | 12.1 ± 0.6 | 246.2 ± 18.7 |

### 3.3. Corrosion Resistance Evaluation

The electrochemical behavior of the Ni–Ru–P coatings in both as-deposited and annealed states were analyzed through potentiodynamic polarization test and corroded surface images. Figure 4 indicates the polarization curves of the as-deposited Ni–Ru–P coatings. The calculated $E_{corr}$ and $I_{corr}$ values for the all the coatings are summarized in Table 3. The A, Ni$_{78.9}$Ru$_{3.3}$P$_{17.8}$ and B, Ni$_{72.0}$Ru$_{12.2}$P$_{15.8}$ coatings show relatively higher $E_{corr}$ values than C, Ni$_{58.5}$Ru$_{27.3}$P$_{14.2}$ film in the as-deposited condition.

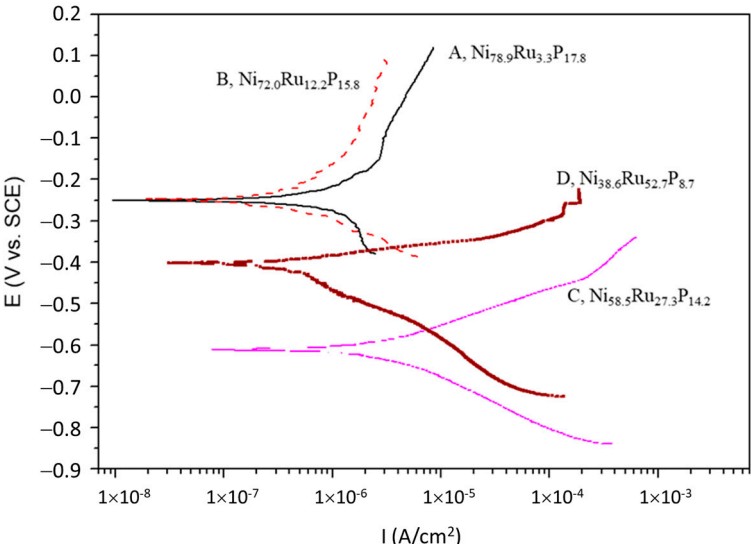

**Figure 4.** The potentiodynamic polarization curves of the as-deposited Ni–Ru–P coatings.

**Table 3.** The $E_{corr}$ and $I_{corr}$ of the Ni–Ru–P coatings in as-deposited and annealed states.

| Coating Systems | Process Condition | $E_{corr}$ (V vs. SCE) | $I_{corr}$ (Amps/cm$^2$) |
|---|---|---|---|
| A, Ni$_{78.9}$Ru$_{3.3}$P$_{17.8}$ | 200 °C as-deposited | −0.25 | $6.04 \times 10^{-6}$ |
| | 475 °C annealed | −0.54 | $3.21 \times 10^{-6}$ |
| | 550 °C annealed | −0.55 | $3.55 \times 10^{-6}$ |
| B, Ni$_{72.0}$Ru$_{12.2}$P$_{15.8}$ | 200 °C as-deposited | −0.25 | $2.22 \times 10^{-6}$ |
| | 475 °C annealed | −0.54 | $4.22 \times 10^{-6}$ |
| | 550 °C annealed | −0.54 | $4.56 \times 10^{-6}$ |
| C, Ni$_{58.5}$Ru$_{27.3}$P$_{14.2}$ | 200 °C as-deposited | −0.61 | $2.96 \times 10^{-6}$ |
| | 475 °C annealed | −0.55 | $5.61 \times 10^{-6}$ |
| | 550 °C annealed | −0.53 | $8.44 \times 10^{-6}$ |
| D, Ni$_{38.6}$Ru$_{52.7}$P$_{8.7}$ | 200 °C as-deposited | −0.40 | $3.04 \times 10^{-6}$ |
| | 550 °C annealed | −0.59 | $6.95 \times 10^{-6}$ |

It seems that the variation in Ru content is responsible for the changes in $E_{corr}$ values between sample A, B, and C. Yet P content would be the key to this issue. With high P content over 9 at.%,

the Ni–P-based alloy coatings tend to be nanocrystalline. A higher P content may further induce an amorphous structure for the Ni–P-based coatings. Since those coatings in this study possess a major Ni amorphous/nanocrystalline microstructure, the decrease in P content in the coatings is responsible for the degeneration of $E_{corr}$ for samples A, B, and C. When the Ru content further increases to 52.7 at.% in the Ni–Ru–P coatings, the $E_{corr}$ is elevated to −0.40 V (which is still more active than A and B coatings). It should be noted that the crystallized Ni, Ru, and Ru$_2$P phases are present for the Ni$_{38.6}$Ru$_{52.7}$P$_{8.7}$ coating. Another fact is that the chemically inert Ru dominates in the Ni$_{38.6}$Ru$_{52.7}$P$_{8.7}$ coating. Moreover, it has been demonstrated that a medium P content around 8 at.% is beneficial for better corrosion resistance in Ni–P alloy system [27]. Thus a more positive potential for the Ni$_{38.6}$Ru$_{52.7}$P$_{8.7}$ film is expected as compared to the Ni$_{58.5}$Ru$_{27.3}$P$_{14.2}$ coating.

After annealing processes at 475 or 550 °C for 3 h, the $E_{corr}$ values for all coatings drop to −0.59 to −0.53 V. The corrosion current density for all the coatings keeps at an order of 10$^{-6}$ Amp/cm$^2$. The degeneration of corrosion potential in coatings A and B is significant. This is due to the phase transformation of the amorphous/nanocrystalline structure to a crystallized Ni(Ru) with Ni$_x$P$_y$ compounds, as illustrated in Figures 1 and 2. The ternary amorphous Ni–P-based coating exhibited a higher corrosion resistance than its crystallized form. The crystallized structure provides better opportunities for corrosion solution to go along the grain boundaries and defects. A lower corrosion resistance is thus reasonable for the annealed Ni–Ru–P films. Nevertheless, it has been proved that the Ni–P and its composite coatings present the a relatively noble corrosion potential due to microstructure change when annealed at 400 and 500 °C [14]. In the present study, the $I_{corr}$ values deduced for the annealed samples are within the same order of 10$^{-6}$ Amp/cm$^2$ for as-deposited ones, implying the corrosion currents for the Ni–Ru–P films do not increase significantly after heat treatment.

The corroded surface images were analyzed to determine the corrosion mechanism for the Ni–Ru–P coatings. The as-deposited Ni–Ru–P films remain a smooth and dense feature, similar to that in Figure 3a, after corrosion in 3.5 M NaCl solution. On the other hand, the pitting corrosion behavior is found for the 475 and 550 °C annealed coatings, as illustrated in Figure 5. For 475 °C annealed C, Ni$_{58.5}$Ru$_{27.3}$P$_{14.2}$ coating, etching pits with diameter of approximately 40 to 50 nm are observed. The pitting corrosion phenomenon is also found for the 550 °C annealed D, Ni$_{38.6}$Ru$_{52.7}$P$_{8.7}$ coating. Intense pitting corrosion with large pit size over 100 nm is shown for annealed coatings with less Ru contents, as illustrated in Figure 5c. The explanation can be two-fold. Firstly the structure of the low Ru content coatings comprises of dominated Ni phase and Ni$_x$P$_y$ compounds. The crystallized Ni microstructure is more easily to be corroded away since more grain boundaries are present after annealing. Secondly, lower Ru content decreases the chemical inertness of the Ni–Ru–P films intrinsically. As a consequence, the annealed coatings with less Ru content, like A, Ni$_{78.9}$Ru$_{3.3}$P$_{17.8}$ and B, Ni$_{72.0}$Ru$_{12.2}$P$_{15.8}$ films, exhibit severe pitting and significant drop in corrosion resistance.

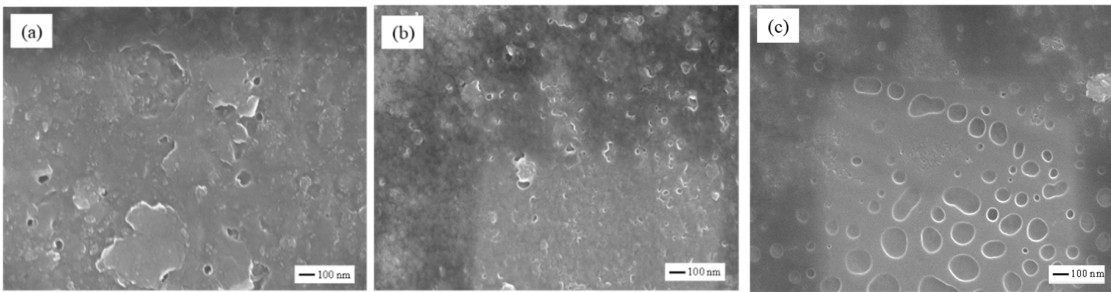

**Figure 5.** The FE-SEM surface images of corroded (**a**) 475 °C annealed C, Ni$_{58.5}$Ru$_{27.3}$P$_{14.2}$, (**b**) 475 °C annealed D, Ni$_{38.6}$Ru$_{52.7}$P$_{8.7}$ and (**c**) 550 °C annealed B, Ni$_{72.0}$Ru$_{12.2}$P$_{15.8}$ coatings.

## 4. Conclusions

Ni–Ru–P alloy coatings with Ru contents from 3.3 to 52.7 at.% are fabricated by magnetron dual-gun co-sputtering technique with target input power control. The as-deposited Ni–Ru–P coatings with Ru

concentration below 27.3 at.% show an amorphous/nanocrystalline microstructure. The high Ru-content $Ni_{38.6}Ru_{52.7}P_{8.7}$ coating, on the other hand, exhibits crystallized Ni, Ru, and $Ru_2P$ mixed phases in its as-fabricated state. The amorphous/nanocrystalline Ni–Ru–P thin films transform into Ni(Ru) and $Ni_xP_y$ crystallized phases under annealing, while the high Ru-content $Ni_{38.6}Ru_{52.7}P_{8.7}$ coating possesses a stable mixed-phase structure against heat treatment. The as-fabricated Ni–Ru–P films show an increased hardness from 7.2 to 10.4 GPa with the Ru contents. Due to phase transformation, the annealed films possess an increased surface hardness. The highest film hardness of 12.1 GPa is found for the 550 °C heat-treated high Ru-content $Ni_{38.6}Ru_{52.7}P_{8.7}$ coating owing to its stable mixed-phase structure. For the as-deposited coatings with an amorphous/nanocrystalline feature, the decrease in P content is responsible for the decrease in corrosion potential. The degeneration of $E_{corr}$ for the annealed Ni–Ru–P coatings is attributed to the crystallized phases of Ni and $Ni_xP_y$ compounds. All the annealed Ni–Ru–P coatings show a pitting corrosion feature in a 3.5 M NaCl solution. Severe localized pitting on 550 °C annealed low Ru content Ni–Ru–P coatings is observed. The development of Ni(Ru) and $Ni_xP_y$ crystallized phases during annealing is responsible for the degeneration of corrosion resistance.

**Author Contributions:** Conceptualization, F.-B.W.; methodology, Y.-C.H.; investigation, Y.-C.H.; data curation, Y.-C.H.; writing—original draft preparation, Y.-C.H.; writing—review and editing, F.-B.W.; visualization, Y.-C.H.; supervision, F.-B.W.; funding acquisition, F.-B.W. All authors have read and agreed to the published version of the manuscript.

**Funding:** This research was funded by Ministry of Science and Technology, Taiwan, under contracts Nos. NSC-98-2221-E-239-009-MY3, MOST-107-2221-E-239-004-MY3, MOST-107-2218-E-011-017 and MOST-108-2218-E-011-008.

**Acknowledgments:** Technical support and constructive discussion from S.Y. Tsai and J.G. Duh in the Field-Emission Electron Probe Microanalysis, FE-EPMA, Lab, Instrumental Center, National Tsing Hua University are highly acknowledged.

**Conflicts of Interest:** The authors declare no conflict of interest.

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
