# Peer review of "Thermal Stability, Hardness, and Corrosion Behavior of the Nickel–Ruthenium–Phosphorus Sputtering Coatings"

_coatings, doi:10.3390/coatings10080786_

Round 1

Reviewer 1 Report

1.Introduction of a noble metal as part of the coating alloy should be justified in terms of industrial importance (cost vs high quality coating) in one (max. two) lines in the introduction. Otherwise the work will remain as pure academic.

2.Hardness of the coatings may be useful to be mentioned in parallel with the corrosion resistance. 

3. Please mention the sputtering method (DC or RF). Ni sputtering is not trivial.

4. Please emphasize the original ideas of the work  - is it only the Ru adding into the coating.

Reviewer 2 Report

The title of the paper sounds interesting as it focused on the thermal stability and corrosion behavior of Ni-Ru-P sputtering coatings. Especially the annealed part of the work interest me with its corrosion behavior. 

My main question is what's new in this work, do some works in this area already published, I could not find this section. Although authors mention that there are binary compounds such as Ni-P based coating already discussed. so, this is the first attempt in terms of Ru ternary addition.

Second Fig. 4 needs to redrawn, it's not clear.

Third regarding references authors need to include some of the related 

  1. High-temperature oxidation of metals ( how the metals and then alloy behave in high temperature in annealing condition).
